# Adaptive deep brain stimulation as advanced Parkinson's disease treatment (ADAPT study): protocol for a pseudo-randomised clinical study

Dan Piña-Fuentes,[1] Martijn Beudel,[2,3] Simon Little,[4] Peter Brown,[5,6] D L Marinus Oterdoom,[1] J Marc C van Dijk[1]

For numbered affiliations see end of article.

**Correspondence to**
Mr. Dan Piña-Fuentes;
d.a.i.pina.fuentes@umcg.nl,
dr.danfuentes@gmail.com

## ABSTRACT

**Introduction** Adaptive deep brain stimulation (aDBS), based on the detection of increased beta oscillations in the subthalamic nucleus (STN), has been assessed in patients with Parkinson's disease (PD) during the immediate postoperative setting. In these studies, aDBS was shown to be at least as effective as conventional DBS (cDBS), while stimulation time and side effects were reduced. However, the effect of aDBS on motor symptoms and stimulation-induced side effects during the chronically implanted phase (after the stun effect of DBS placement has disappeared) has not yet been determined.

**Methods and analysis** This protocol describes a single-centre clinical study in which aDBS will be tested in 12 patients with PD undergoing battery replacement, with electrodes implanted in the STN, and as a proof of concept in the internal globus pallidus. Patients included will be allocated in a pseudo-randomised fashion to a three-condition (no stimulation/cDBS/aDBS), cross-over design. A battery of tests will be conducted and recorded during each condition, which aim to measure the severity of motor symptoms and side effects. These tests include a tablet-based tapping test, a subscale of the Movement Disorder Society-unified Parkinson's disease rating scale (subMDS-UPDRS), the Speech Intelligibility Test (SIT) and a tablet-based version of the Stroop test. SubMDS-UPDRS and SIT recordings will be blindly assessed by independent raters. Data will be analysed using a linear mixed-effects model.

**Ethics and dissemination** This protocol was approved by the Ethical Committee of the University Medical Centre Groningen, where the study will be carried out. Data management and compliance to research policies and standards of our centre, including data privacy, storage and veracity, will be controlled by an independent monitor. All the scientific findings derived from this protocol are aimed to be made public through publication of articles in international journals.

**Trial registration number** NTR 5456; Pre-results.

## INTRODUCTION
### Background and rationale

Deep brain stimulation (DBS) is currently one of the most effective advanced treatments

### Strengths and limitations of this study

► This protocol aims to assess both efficacy and side-effect profile of adaptive deep brain stimulation.
► The stun effect will not present in the patient population to be included in this protocol.
► Intraoperative testing will minimise exposure time for electrodes and risk of infection.
► Testing and washout periods will be limited by the intraoperative time available.
► The intraoperative setting will limit the tests available to be assessed in patients with Parkinson's disease.

for patients with Parkinson's disease (PD). At present, conventional DBS (cDBS) provides the continuous delivery of fixed-frequency and fixed-amplitude electrical pulses through the electrodes implanted mostly in either the subthalamic nucleus (STN) or the internal globus pallidus (GPi).[1 2] Although cDBS is an effective treatment, it still has limitations in terms of efficacy, side effects and efficiency, and these have not undergone substantial improvements since the implementation of cDBS in clinical practice, approximately 20 years ago.[3] However, new DBS systems directed to improve current stimulation algorithms are currently under development. These systems are often referred to as adaptive DBS (aDBS), as they aim to modulate the delivery of electrical stimulation, controlled by the detection of symptoms, such as tremor, or aberrant signals (biomarkers) that have been implicated in the pathophysiology of the disease.[4] The ultimate goal of aDBS systems is to reduce the total amount of stimulation delivered, in a way in which beneficial effects of DBS are enhanced, while side effects are reduced.[5] Furthermore, decreasing the total amount of stimulation delivered will allow to extend the battery life (span) of DBS devices.

Several biomarkers in PD have been proposed for use as a feedback signal for aDBS.[6] Currently, the presence of prominent beta oscillations (13–35 Hz), in either the STN or the GPi, seems to be the most promising since it correlates with the severity of bradykinesia and rigidity.[7] Evidence to date suggests that aDBS may be at least as effective as cDBS systems, while stimulation time and side effects are reduced.[8–10] However, virtually all aDBS experiments based on beta oscillations have thus far only been tested in the immediate postoperative setting. Although aDBS has been evaluated mostly with regard to the STN, preliminary evidence indicates that aDBS could be equally effective when applied to the GPi.[11] The correlation of other prominent PD symptoms (such as tremor) with beta oscillations is far less robust. Therefore, the impact of aDBS on these symptoms is yet to be addressed.[12 13]

Besides its effect on motor symptoms, a common side effect that could potentially be addressed by aDBS is stimulation-induced dysarthria. Preliminary data in the immediate postoperative phase showed that aDBS did not induce dysarthria, in contrast to cDBS.[14] Likewise, due to the reduction in the total amount of stimulation that occurs in aDBS, this new technology could potentially help to tackle DBS-related side effects in executive functions, such as impaired response inhibition.[15 16] Patients with PD treated with DBS are prone to make more mistakes in inhibitory control tests (eg, the Stroop task).[17] Response inhibition is a fundamental cognitive function, and its impairment could lead to impulsive conduct.[18] For that reason, it is possible that stimulation-related impulsivity is the result of an impairment of response inhibition caused by DBS.[19 20] It was first proposed that this side effect was due to unintended stimulation of limbic areas of the STN.[21] However, it has been shown recently that DBS has a positive effect on risk-reward decision behaviour in patients with PD, and that such an effect was mostly dependent on stimulation of the motor part of the STN.[22] This finding indicates that at least some of the non-motor effects of DBS are mediated through direct stimulation of the motor part of the STN. This is supported by increasing evidence that indicates that the motor and non-motor functions of the STN are not perfectly segregated.[23] It has been shown that beta oscillations in the motor STN play a significant role in adequate response inhibition.[24] For that reason, stimulation-induced impulsivity could be the result of an excessive beta suppression produced by continuous stimulation. Therefore, aDBS might be used as an additional option to address stimulation-induced impulsivity, since aDBS has the potential to provide a functional selectivity, by only triggering stimulation based on (primarily) motor biomarkers (eg, beta oscillations), only when a threshold is exceeded. As a result, only excessive beta activity would be suppressed.

## Objectives

The aim of this study is to assess whether aDBS provides a similar motor improvement compared with cDBS, while limiting the presence of stimulation-induced side effects in patients with chronically implanted electrodes. Stimulation efficacy is assessed using a battery of tests aiming to measure the motor response to different conditions and the severity of dysarthria and/or impulsivity. The goals of this protocol are summarised in three questions:

1. Is aDBS at least as efficacious as cDBS in terms of motor improvement?
2. Is the speech-related and response inhibition side-effect profile of aDBS lower than that of cDBS?
3. What is the amount of stimulation provided in aDBS compared with cDBS?

### Trial design

This protocol describes an exploratory study in which patients with PD with DBS electrodes implanted in the STN are allocated to a three-condition (no stimulation (NoStim)/cDBS/aDBS), cross-over design (figure 1). The conditions are pseudo-randomised in a 2:1:1 format, in such a way that for every two patients who start with no-stimulation, one patient will receive aDBS first and one will have cDBS first. The main reason for this proposed pseudo-randomisation is to minimise the impact that an incomplete washout effect derived from stimulation could have on the scores of NoStim condition, as this condition is the baseline for the comparison of the effect of the two stimulation modalities. In four patients, stimulation conditions will be separated by a NoStim condition to minimise the carryover effect that cDBS could have on aDBS (or vice versa). Lastly, NoStim will be the last condition in two patients in order to partially balance the effects of training or fatigue on the tasks. Additionally, pilot data could be collected from two patients in order to explore the effects of aDBS on the GPi.

The performance of several tasks during the three conditions will take place during the battery replacement surgery, when a temporal externalisation of the DBS extension wires is performed. This means that subjects to be included have had their DBS electrodes implanted for at least 3–5 years, and therefore the stun effect is not present in this patient population. It was decided to perform this protocol intraoperatively as it allows the surgery to be completed in one session, minimising the time that electrodes are externalised and patients are exposed to potential risks of infection.

## METHODS AND ANALYSIS
### Methods: participants, interventions and outcomes
#### Study setting

This is a single-centre study at the University Medical Centre Groningen.

#### Inclusion criteria

► Patients with PD with DBS electrodes implanted bilaterally in the STN. A maximum of two patients with PD with electrodes implanted in the GPi would be allowed as pilot data.

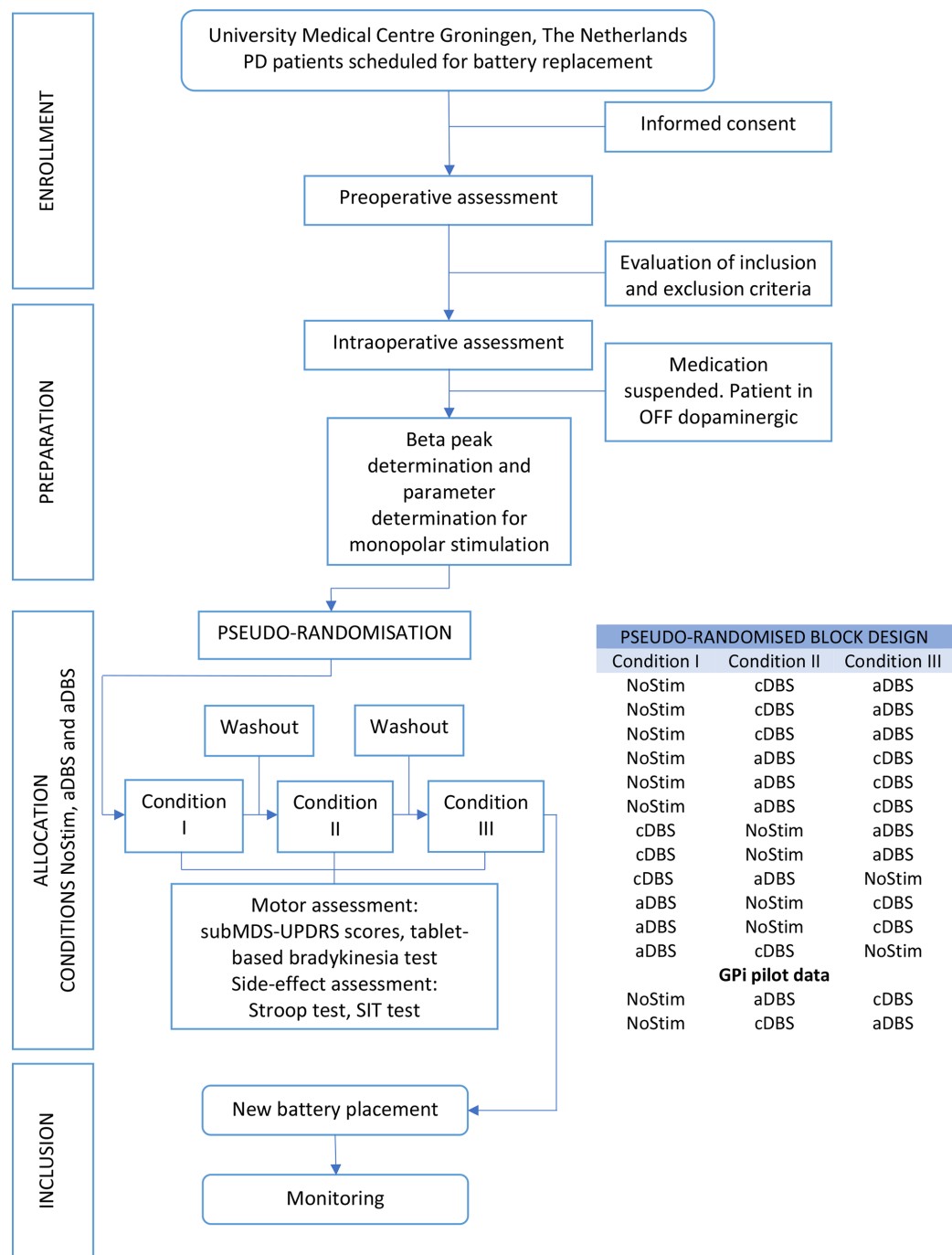

**Figure 1** Trial design and pseudo-randomisation block. aDBS, adaptive deep brain stimulation; cDBS, conventional DBS; GPi, internal globus pallidus; PD, Parkinson's disease; subMDS-UPDRS, subscale of the Movement Disorder Society-unified Parkinson's disease rating scale; SIT, Speech Intelligibility Test.

► Patients are cognitively capable of performing the tests included in this protocol. This item is subjectively evaluated during the patient interview, by looking at the performance of the candidate when completing a trial set of the tablet-based tests used during the experiment.

► Able to provide written informed consent.

► Physical condition that enables 20–30 min of testing. This is evaluated according to the clinical records and the fragility of the patient (determined by the treating neurologist).

► Ability to undergo testing in the OFF dopaminergic state, determined by the clinical condition of the patient after medication withdrawal.

### Exclusion criteria

► Patients who cannot tolerate battery replacement under local anaesthesia (eg, presence of severe OFF medication tremor, severe anxiety, etc).

► <18 years old.

► Clinical disorders or diseases that could interfere with clinical assessment (eg, vision impairment that

cannot be corrected with glasses, severe limb movement restriction).

► Cognitive impairment sufficient to prevent the patient to consider and retain the necessary information to make a decision regarding the participation in the study.

## Investigational device

The responsive (closed loop) aDBS is delivered through a custom-made stimulation-amplifier which allows local field potentials (LFPs) to be recorded during the application of DBS. With this device, LFPs are filtered in real time around a subject's specific beta frequency (±3 Hz around peak).[9] Afterwards, a median threshold is selected in order to deliver monopolar stimulation every time that beta activity in the rectified filtered LFP exceeds the predefined threshold. Stimulation pulses are charge-balanced and asymmetric, mirroring the waveform of the standard clinical Medtronic Activa PC. Pulses are delivered at 130 Hz with a pulse width of 60 µs (standard clinical parameters) using a ramping period of 250 ms at onset and offset in order to help limit the presence of paraesthesias, autonomic or other non-specific side effects, such as dizziness or nausea.[25] From the four contacts available for each electrode lead, two electrodes are required to perform bipolar LFP recordings (either contacts 0–2 or 1–3), while one electrode (in between the two recording electrodes) is left available for stimulation (either contact 1 or 2). The stimulation-amplifier has been built to strict safety guidelines (Report 90 BS EN60601-1 safe design, construction of medical equipment) and extensively tested,[8 9 11 26] and found to be safe and reliable.

## Procedure

Patients have to skip at least one medication dose and are not allowed to be under the effect of prolonged-release dopaminergic medication in order to be classified as OFF medication before surgery. Patients are brought to the operating theatre for their battery replacement procedure. After the battery is exposed, the old battery is detached from the DBS electrodes and explanted. At this moment, two temporary wires are attached to the DBS electrode extension cables and connected to a combined stimulator and amplifier (see previous section). A brief rest recording of 30–60 s is performed from each of the two bipolar montages (0–2 or 1–3) available for each hemisphere. Power spectral density estimates are obtained from each rest recording, and the bipolar contact pair (0–2 or 1–3) per hemisphere with the highest beta peak power is selected. Monopolar cDBS is then titrated using the available contact located between the selected recording contacts. The voltage for monopolar stimulation is titrated initially based on an approximation of the voltage used in clinical practice for each individual patient (independently of the montage used in clinical practice). Voltage is increased or decreased according to the presence of side effects, clinical effect and interference with LFP recordings (in practice, we have seen that

voltages>3.5 V are more prone to cause interference with the recordings, and these are also rare stimulation parameters in clinical practice). Once an optimum stimulation voltage is established, the patients are assigned to each of three counterbalanced conditions in which the patient is either stimulated with aDBS or cDBS, or not stimulated, separated by washout periods of 1–3 min, according to the remaining time available. Stimulation parameters (voltage, frequency, pulse width and contact selection) remain constant for both aDBS and cDBS, differing only in the timing of stimulation delivered according to beta detection. As battery replacements are performed under local anaesthesia, we aim to limit the duration of the experiment to a maximum of 25–30 min.

## Clinical tests and outcomes

Each patient performs the following tasks in each of the three pseudo-randomised conditions:

a. Tablet-based tapping test: several analogue and digital versions of the tapping test have been validated to objectively measure the presence of bradykinesia in patients with PD.[27–29] We developed a custom tablet-based finger tapping test using the PsyPad platform.[30] In this test, a tablet is presented to the patient, in which two squares appear at the left and right sides of the screen (figure 2). The patient then has to press the squares in an alternating pattern. We tested this set-up, as a proof of principle, on a patient with PD who underwent battery replacement,[26] using 50 tapping iterations per condition. In order to optimise the time available for the battery of tests, it was decided to reduce the iterations to 20. Since the duration of each iteration was approximately 500 ms, 20 iterations (10 back-and-forth screen tapings) will be roughly equivalent to 10 finger tapings, which is the amount of taps required to measure tapping items from the Movement Disorder Society-unified Parkinson's disease rating scale (MDS-UPDRS).

b. Subscale of the MDS-UPDRS (subMDS-UPDRS)[31] (Part III, items 3.4, 3.5, 3.6, 3.15, 3.17 a, b): as patients are lying on the operating table during the experiment, only selected items can be performed by the patients. The items that are included and filmed are finger tapping, hand movements, pronation-supination, rest tremor and postural tremor.

c. Speech Intelligibility Test (SIT): this is a modified version of the Assessment of Intelligibility for the Dysarthric Speech.[32 33] In this test, the patient is instructed to read out a list of 18 sentences (approximately 110 words per condition), while the speech is recorded with a mobile application. The words that form each sentence are randomly chosen, taking into account that each sentence must follow the correct order of a proper grammatical construction. Voice recordings are sent to a speech therapist, who blindly evaluates the sentences and writes them down. Afterwards, the sentences written by the speech therapist are compared with the

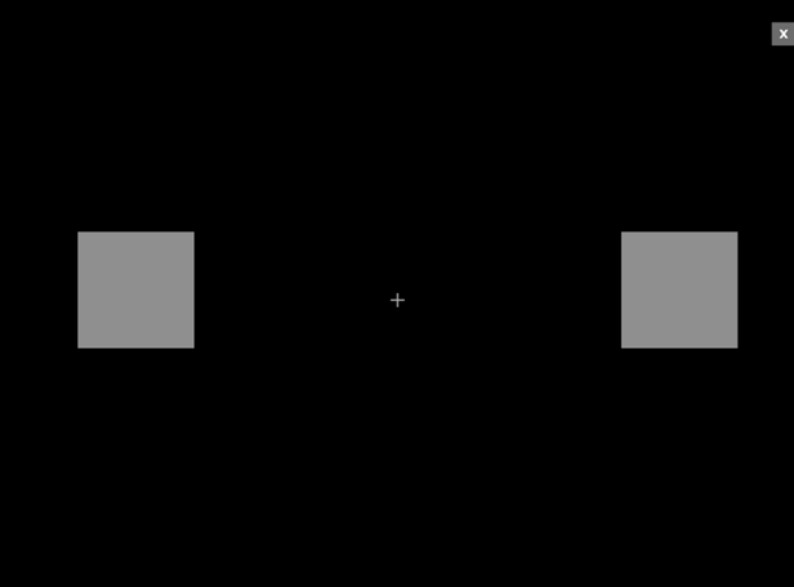

**Figure 2** Example of a trial of the tablet-based bradykinesia test.

original list, and the amount of words that coincide are expressed as a percentage of the total words.

d. Tablet-based version of the Stroop test[34]: in this test, 20 colour words are presented to each patient in a sequential order, and each word is coloured as shown in figure 3. From those 20 words, five are coloured in the same colour as the word written (congruent), while 15 words are coloured in a colour different from the word written (incongruent). We chose an unbalanced design as our main focus is the effect of stimulation on incongruent words between conditions, rather than the effect of stimulation between congruent and incongruent words. For each word presented, four buttons are presented at the bottom of the tablet screen, each containing different colour words. The patient has to select the button containing the colour word presented on the screen, independently of the colour in which the word is coloured in. Reaction times and accuracy for each trial are directly uploaded to an online database, from which they can be extracted for data analysis.

### Order of the tests

After the battery is exposed, and before the performance of the clinical tests can start, a number of preparations are required in order to programme the aDBS (see the Procedure section). This requirement can lead to individual differences in the time available to perform the clinical tests due to the maximum amount of intraoperative time allowed for this experiment. Therefore, the sequence in which the clinical tests are performed on each condition is initially ordered according to the approximate time it takes the patient to perform each of them, the suggested order being as follows: (1) tablet-based tapping test, (2) subMDS-UPDRS, (3) Speech and Intelligibility Test (SIT), (4) tablet-based version of the Stroop test. If during the course of each individual experiment it is considered that the time will not be sufficient for a patient to complete the full protocol, only the first test(s) could be selected to be performed in order to ensure that at least one of the tests will be completed for each of the three conditions.

### Sample size

According to previous studies,[35] a difference of approximately four points in the total motor UPDRS could be considered clinically significant (3.7% of the total scale). This would be roughly equivalent to a limit of 1.5 points on a 10-item UPDRS scale, which is the one used in this protocol. Based on pilot data conducted in Oxford/London on four patients,[9] aDBS scores had an standard

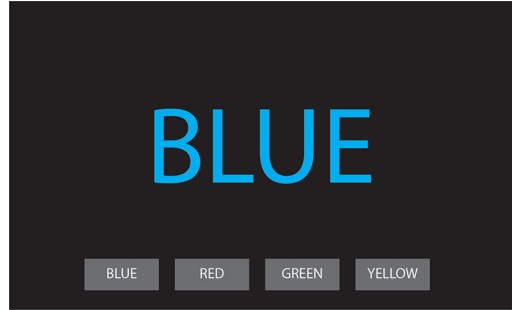
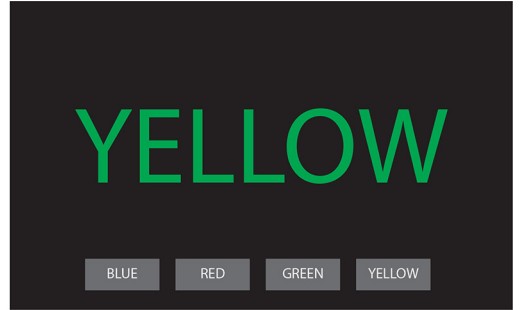

**Figure 3** Example of two trials of the tablet-based Stroop test with congruent (left) and incongruent (right) conditions.

error of approximately two points (SD of 4). This is roughly equivalent to an SD of 1.5 points on a 10-item UPDRS scale. Using an α of 0.05, a power (1−β) of 0.8 and assuming similar variances, a sample size of 12 patients would be necessary to demonstrate that the efficacy of aDBS is not inferior to cDBS, while providing the potential benefits of facilitating programming, battery life optimisation and the potential to reduce stimulation-induced side effects. A potential extension of two patients (14 patients in total) would be pursued in order to explore the potential effects of aDBS on patients with electrodes implanted on the GPi.

## Recruitment

Patients who are scheduled for a battery replacement at the University Medical Centre Groningen are contacted by telephone 2 weeks before their hospital admission to be informed about the study. Complementary written information is sent by mail to patients who show an interest in participating in the study. At the time of hospital admission, each patient is personally approached by one of the members of the research team, if they meet all of the inclusion criteria and none of the exclusion criteria. Patients who decide to participate in the study will then be given the opportunity to ask further questions and invited to sign the consent form.

## Methods: assignment of interventions
### Allocation

The ultimate goal of the allocation procedure is to have a pseudo-randomisation in which all the possible combinations of the block design are covered (ie, 2:1:1, see the Trial design section, figure 1). For that reason, patients will be sequentially allocated to each possible pseudo-randomisation scenario until 14 patients are allocated. Only successful bilateral measurements will be considered in the pseudo-randomisation list. At first, a pseudo-randomisation of 3:1:1 (10 patients) will be set as the allocation goal in order to ensure that all conditions of the pseudo-randomisation could be met by mid-2019. Depending on the number of aborted experiments and the deadline for inclusion, two patients more may potentially be included to achieve the 2:1:1 pseudo-randomisation.

## Blinding (masking)

Patients will be blinded to the stimulation condition. Video and voice recordings will be taken for the subMDS-UPDRS and SIT parts, and will be blindly evaluated by two neurologists experienced in movement disorders and a speech therapist, respectively. Analysis of tablet-based outcomes will be automated and matched across the three conditions.

## Methods: data collection, management and analysis
### Data management

Data are stored on a central database using REDCap storage system, being accessible only to the authorised personnel involved in this study.

## Statistical methods

Simple scores from the subMDS-UPDRS, tablet-based tapping test, SIT and Stroop test during aDBS versus cDBS or NoStim will be checked for normality using Q-Q plots, and if normally distributed, will be compared using a linear mixed-effects model. If the data are non-normally distributed, a data transformation will be applied, or a distribution different from normal will be employed for the test.

## Methods: monitoring
### Harms

The experimental procedure does not substantially expose patients to increased risks, as there are no extra surgical procedures required, and patients are stimulated only with less electrical current than in their conventional setting. Given that the experiments take place in the operation room, and only add less than 30 min to the total surgery duration, there is no significant added risk of infection. Since patients are tested in the OFF dopaminergic state, this exaggerates their symptoms to certain extent. However, DBS remains active until the moment of the surgery, which limits the impact of medication discontinuation. In previous aDBS studies, electrodes were exposed for hours or days[9 10] without any increased risk of infection. Therefore, as battery replacement is completed in one session, the amount of time in which the extension wires are exposed is considerably shortened in this study. Nevertheless, infections will be considered an adverse event and infection rates monitored.

### Monitoring

Data management and compliance to research policies and standards of our centre, including data privacy, storage and veracity, are verified by an independent monitor from a different department within the hospital. As this is an investigator-initiated study, the project leader submits, once a year throughout the clinical trial, a safety report to the accredited local ethics committee. Serious adverse events, defined by the Central Committee on Research Involving Human Subjects in the Netherlands as 'any untoward medical occurrence or effect that at any dose results in death, is life threatening (at the time of the event), requires hospitalisation or prolongation of existing inpatients' hospitalisation or results in persistent or significant disability or incapacity', will be promptly reported to the corresponding governmental authorities.

### Patient and public involvement

In order to make sure that the experiments are as comfortable as possible for the subjects, a short Patient Reported Outcome (PRO) questionnaire will be conducted on the first five patients (figure 4). This PRO consists of five questions in which patients can indicate whether the communication was adequate and empathetic and whether the procedure was not too demanding, uncomfortable or painful. The feedback of these reports will be used to evaluate the adequacy of the protocol, and changes will

**Patient Reported Outcome Form**

1. Are you satisfied with the information you received before the experiment?

Yes ☐   No ☐

If not, could you indicate which kind of information would you have liked to receive regarding the experiment?

_______________________________________

_______________________________________

2. Was the experiment burdensome for you?

Yes ☐   No ☐

If this was the case, could you please specify which part of the experiment was burdensome for you?

_______________________________________

_______________________________________

3. Did you feel uncomfortable during the experiment?

Yes ☐   No ☐

If yes, could you please specify which part of the experiment made you feel uncomfortable?

_______________________________________

_______________________________________

4. Do you consider that the communication during the performance of the experiment and/or about what you could expect during the experiment was adequate?

Yes ☐   No ☐

If this was not the case, could you please specify in which part of the experiment the communication can be improved?

_______________________________________

_______________________________________

5. Did you experience any form of pain during the experiment?

Yes ☐   No ☐

If this was the case, could you please specify in which part of the operation you experienced pain?

_______________________________________

_______________________________________

**Figure 4**   Example of the Patient Reported Outcome questionnaire form conducted on the participants.

be incorporated into the following study procedures, if necessary via an amendment.

## ETHICS AND DISSEMINATION
### Protocol amendments
The protocol presented here is the fifth version (9 December 2018) of the original protocol prepared for this study. The second version of the protocol (15 September 2015) got the approval of the ethical committee after several modifications of the first version (22 July 2015). In version 3 (12 April 2017), an attempt was made to include patients undergoing new implantations. However, this was dropped in subsequent versions, since (a) patients undergoing new implantations were often considerably fatigued at the moment of the measurement and could not perform all the tasks; (b) differences in total intraoperative time between new implantations

and battery replacements meant that the time available for the measurement was considerably less during new implantations; (c) the stereotactic frame slightly restricted the visual field of the patients in new implantations. Version 4 (7 September 2017) and version 5 were meant to increase the total number of patients to be included, as several failed or incomplete measurements took place during the initial inclusions.

## Consent

During the inclusion interview, patients are informed that the participation in this study is on an entirely voluntary basis, and that they are therefore allowed at any moment to withdraw their consent. No economic reward is provided for the participation in this study.

## Confidentiality

Only indirect personal data from each patient are stored as coded information in order to prevent the possibility that patient data could be traced back to an individual patient. The coding list will be safeguarded in a secured facility located in our centre. Data derived from the clinical tests or recordings could be exported only as anonymised data to be used for statistical analysis.

## Ancillary and post-trial care

Patients are insured up to €650 000 for harm suffered directly from trial participation. Details of the insurance are provided on the patient information letter of this study.

**Author affiliations**
[1]Department of Neurosurgery, University Medical Centre Groningen, Groningen, The Netherlands
[2]Department of Neurology, University Medical Centre Groningen, Groningen, The Netherlands
[3]Department of Neurology, Amsterdam University Medical Centre, The Netherlands
[4]Sobell Department of Motor Neuroscience and Movement Disorders, Institute of Neurology, Queen Square, London, UK
[5]Medical Research Council Brain Network Dynamics Unit, University of Oxford, Oxford, UK
[6]Nuffield Department of Clinical Neurosciences, University of Oxford, United Kingdom

**Contributors** MB and JMCvD designed and prepared the first draft of the protocol, and initiated the study. DP-F formatted the protocol according to the Standard Protocol Items: Recommendations for Interventional Trials (SPIRIT) guidelines and prepared the manuscript. SL, PB and DLMO critically reviewed the protocol procedures and manuscript. All authors reviewed and approved the final version of this manuscript.

**Funding** This study is publicly funded by grants received from the Dutch Brain Foundation ('Hersenstichting Nederland', grant F2015(1)-04), the National Mexican Council of Science and Technology (CONACYT) grant number CVU 652927 and the University of Groningen/University Medical Centre Groningen (RuG/UMCG). SL is personally supported by a Wellcome trust postdoctoral clinical research training grant (105804/Z/14/Z).

**Disclaimer** All the scientific findings derived from this protocol are aimed to be made public through publication of articles in international journals.

**Competing interests** None declared.

**Patient consent for publication** Not required.

**Ethics approval** This protocol is approved by the local ethics committee of our institution, and all procedures are carried out according to the Declaration of Helsinki and the Dutch legislation on medical research involving human subjects.

**Provenance and peer review** Not commissioned; externally peer reviewed.

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
