## [Reviewer comments · BMJ Open]

ARTICLE DETAILS

TITLE (PROVISIONAL)	Adaptive Deep Brain Stimulation as Advanced Parkinson's disease Treatment (ADAPT study): protocol for a pseudo-randomised clinical study.
AUTHORS	Piña-Fuentes, Dan; Beudel, Martijn; Little, Simon; Brown, Peter; Oterdoom, D.L.Marinus; van Dijk, J. Marc C.

VERSION 1 - REVIEW

REVIEWER	Aristide Merola University of Cincinnati, USA
REVIEW RETURNED	11-Feb-2019

GENERAL COMMENTS	This is an interesting protocol regarding the study of adaptive DBS as compared to conventional DBS. The introduction is balanced and the aims are clearly stated. However, this reviewer feels that some issues require additional attention. In particular: 1) Testing adaptive DBS in patients that have already received 3-5 years of previous stimulation has to be discussed as a possible bias rather than a strength of the study. We do not know to what extent 3-5 years of previous non-adaptive stimulation might have influenced brain plasticity and, therefore, to what extent the study findings will be generalizable to a population of DBS-naive PD patients.2) I am not convinced by the sample size calculation. First of all, the sample size calculation has to be based on the primary outcome measure. The authors should either replace the order of the aims or re-calculate the sample size based on motor improvement and provide a rationale to justify that the sample size calculated covers also the other aims. Also, additional details should be provided on previous literature used to calculate the sample size. Finally, the number of 14 patients seems very low. I strongly recommend a statistical revision of the sample size calculation.3) The inclusion of both STN-DBS and GPi-DBS requires additional analysis to estimate the efficacy of adaptive DBS in the two targets, which significantly differ for anatomical and functional connections. Either the authors decide to focus on one target (reasonably STN) or (even better) they increase the sample of patients included in order to guarantee that the study is appropriately powered to demonstrate the effect (and possibly differences) of adaptive DBS in the two separate population of patients. Also, the authors need to consider that the efficacy of DBS in GPi usually requires weeks to fully manifest, differently from STN-DBS. This point has to be carefully addressed.
--

REVIEWER	Atsushi Umemura National Hospital Organization Utano Hospital, Department of Neurology
REVIEW RETURNED	14-Feb-2019

GENERAL COMMENTS	1) In the Introduction section, the authors report that DBS can induce side effects, such as dysarthria and impulse control disorder, and aDBS could potentially help to tackle those non-motor side effects. It could be useful to add to the psychological assessment focusing on impulse control disorder not only the SIT and Stroop test. 2) The words “cognitive side-effect profile (page 4, line 43)” is unclear. Do the authors want to evaluate executive functions using the Stroop Test? 3) As the authors are described, crossover design is need to minimize the impact of incomplete washout-effect. The authors should specify the washout time between the three conditions. 4) According to the inclusion and exclusion criteria, this study protocol allows the candidate in a wide age range. The vulnerability to neuropsychiatric disorders in PD patients varies by age; younger patients are susceptible to be impulse control disorder, and older patients are to be cognitive dysfunctions. Those variances among the participants may obscure the statistical results.
---

VERSION 1 – AUTHOR RESPONSE

Reviewer(s)' Comments to Author:

Reviewer: 1

Reviewer Name: Aristide Merola

Institution and Country: University of Cincinnati, USA

Please state any competing interests or state 'None declared': None declared

Please leave your comments for the authors below

This is an interesting protocol regarding the study of adaptive DBS as compared to conventional DBS. The introduction is balanced and the aims are clearly stated. However, this reviewer feels that some issues require additional attention. In particular:

1) Testing adaptive DBS in patients that have already received 3-5 years of previous stimulation has to be discussed as a possible bias rather than a strength of the study. We do not know to what extent 3-5 years of previous non-adaptive stimulation might have influenced brain plasticity and, therefore, to what extent the study findings will be generalizable to a population of DBS-naïve PD patients.

Adaptive DBS has virtually only been tested on patients in the immediate postoperative setting, i.e. on DBS-naïve patients (Little et al. 2013, 2016; Arlotti et al. 2018), in which it was shown to be at least as effective as conventional DBS, with potential benefits of facilitate (initial) programming, battery-saving properties, and better side-effect profile (Tripoliti et al. 2016). Therefore, our research protocol focuses on patients who already have electrodes implanted for a long time, in order to see whether

beta power is still a reliable biomarker for the clinical status of the patient, and to see how the aforementioned plasticity might have an effect on the response to adaptive DBS. By doing the measurement in the battery replacement phase, we are aiming to complement the knowledge available by addressing this issue, and for that reason we consider that a strength of our protocol.

Little S, Pogosyan A, Neal S, et al. Adaptive Deep Brain Stimulation In Advanced Parkinson Disease. *Ann. Neurol.* 2013;74:449–57. doi:10.1002/ana.23951

Little S, Beudel M, Zrinzo L, et al. Bilateral adaptive deep brain stimulation is effective in Parkinson's disease. *J Neurol Neurosurg Psychiatry* 2016;87:717–21. doi:10.1136/jnnp-2015-310972

Arlozzi M, Marceglia S, Foffani G, et al. Eight-hours adaptive deep brain stimulation in patients with Parkinson disease. *Neurology* 2018;90:e971–6. doi:10.1212/WNL.0000000000005121

Little S, Tripoliti E, Beudel M, et al. Adaptive deep brain stimulation for Parkinson's disease demonstrates reduced speech side effects compared to conventional stimulation in the acute setting. *J. Neurol. Neurosurg. Psychiatry.* 2016;87:1388–9. doi:10.1136/jnnp-2016-313518

2) I am not convinced by the sample size calculation. First of all, the sample size calculation has to be based on the primary outcome measure. The authors should either replace the order of the aims or re-calculate the sample size based on motor improvement and provide a rationale to justify that the sample size calculated covers also the other aims. Also, additional details should be provided on previous literature used to calculate the sample size. Finally, the number of 14 patients seems very low. I strongly recommend a statistical revision of the sample size calculation.

We have recalculated the sample size based on our primary outcome (clinical response in aDBS compared to cDBS), together with references to the literature used for the calculation. As this is a cross-over design, in which we are aiming to demonstrate the non-inferiority of aDBS, compared to cDBS, the number of patients needed is lower than, for example, parallel designs.

3) The inclusion of both STN-DBS and GPi-DBS requires additional analysis to estimate the efficacy of adaptive DBS in the two targets, which significantly differ for anatomical and functional connections. Either the authors decide to focus on one target (reasonably STN) or (even better) they increase the sample of patients included in order to guarantee that the study is appropriately powered to demonstrate the effect (and possibly differences) of adaptive DBS in the two separate population of patients. Also, the authors need to consider that the efficacy of DBS in GPi usually requires weeks to fully manifest, differently from STN-DBS. This point has to be carefully addressed.

We have focused our design primarily in DBS-STN patients, as the majority of the patients coming to this center have been implanted in the STN. We have left an optional space of two patients above the target of our sample size calculation, in order to explore the effects of aDBS on the GPi of PD patients.

Reviewer: 2

Reviewer Name: Atsushi Umemura

Institution and Country: National Hospital Organization Utano Hospital, Department of Neurology

Please state any competing interests or state 'None declared': None declared.

Please leave your comments for the authors below

1) In the Introduction section, the authors report that DBS can induce side effects, such as dysarthria and impulse control disorder, and aDBS could potentially help to tackle those non-motor side effects. It could be useful to add to the psychological assessment focusing on impulse control disorder not only the SIT and Stroop test.

Due to the intraoperative nature of the experiment, and the time constrains that this involves, it is not feasible for us to perform a thorough psychological assessment in order to explore a possible impulse control disorder. Therefore, we have only approached impulsivity as a symptom throughout this protocol. The Stroop test has been used to evaluate the effect of medication (Djamshidian et al. 2011) and DBS (Jahanshahi et at. 2000) on inhibition, in which a reduction on reaction time were demonstrated on both (faster responses compared to no stimulation/no medication). While this effect could in theory be beneficial, the synergy of both treatments could cause the development of impulsivity in predisposed patients.

2) The words “cognitive side-effect profile (page 4, line 43)” is unclear. Do the authors want to evaluate executive functions using the Stroop Test?

We have changed the word to disinhibition side-effect profile, as that is the main function we want to assess with the Stroop test.

3) As the authors are described, crossover design is need to minimize the impact of incomplete washout-effect. The authors should specify the washout time between the three conditions.

This has been added to the protocol. The minimum washout period will be 1 min between conditions, and more time could be allocated to the washout period, depending on the remaining time available between conditions, for each patient. However, as we have stated, we also tried to control for this factor with the randomisation block.

4) According to the inclusion and exclusion criteria, this study protocol allows the candidate in a wide age range. The vulnerability to neuropsychiatric disorders in PD patients varies by age; younger patients are susceptible to be impulse control disorder, and older patients are to be cognitive dysfunctions. Those variances among the participants may obscure the statistical results.

While no age range was established for this experiment (besides patients being older than 18 years old), the population group will very likely be composed mainly by somewhat older patients. This is because we aim to include patients who already have DBS implanted for at least 3-5 year. Next to this, DBS is usually indicated several years after the beginning of the disease (when fluctuations in medication are becoming troublesome). The age range of the population sample will be further narrowed by our exclusion criteria, which exclude patients who present cognitive decline.

VERSION 2 – REVIEW

REVIEWER	Aristide Merola University of Cincinnati, USA
REVIEW RETURNED	09-Mar-2019

GENERAL COMMENTS	The authors nicely addressed all of my comments. I only have a minor request. Please make sure that the abstract accurately reflects modifications applied to the main text, including the selection of STN as a primary target and GPi as an exploratory outcome.
--

REVIEWER	Atsushi Umemura National Hospital Organization Utano Hospital, Department of Neurology, Japan
REVIEW RETURNED	14-Mar-2019

GENERAL COMMENTS	As the authors referred to the following reports, the Stroop test has been used to evaluate the effect of medication (Djamshidian et al. Parkinsonism Relat Disord 2011; 17: 212–214) and DBS (Jahanshahi et al. Brain 2000; 123: 1142–1154) in PD patients. But even now, I do not know about the utility of that test for the evaluation of impulse control disorder. Djamshidian et al. have reported that there was no difference on the performance of Stroop task between non-impulsive and impulsive PD patients. Jahanshahi et al. have assessed the effects of DBS on executive function using that test. Accordingly, the meaning of the 'disinhibition side-effect profile' keeps obscure for me. The authors should fully address this concern.
---

VERSION 2 – AUTHOR RESPONSE

Reviewer(s)' Comments to Author:

Reviewer: 1

Reviewer Name: Aristide Merola

Institution and Country: University of Cincinnati, USA Please state any competing interests or state 'None declared': None

Please leave your comments for the authors below The authors nicely addressed all of my comments. I only have a minor request. Please make sure that the abstract accurately reflects modifications applied to the main text, including the selection of STN as a primary target and GPi as an exploratory outcome.

Abstract has been modified according to the points indicated.

Reviewer: 2

Reviewer Name: Atsushi Umemura

Institution and Country: National Hospital Organization Utano Hospital, Department of Neurology, Japan Please state any competing interests or state 'None declared': None declared

Please leave your comments for the authors below As the authors referred to the following reports, the Stroop test has been used to evaluate the effect of medication (Djamshidian et al. Parkinsonism Relat Disord 2011; 17: 212–214) and DBS (Jahanshahi et al. Brain 2000; 123: 1142–1154) in PD patients. But even now, I do not know about the utility of that test for the evaluation of impulse control disorder. Djamshidian et al. have reported that there was no difference on the performance of Stroop task between non-impulsive and impulsive PD patients. Jahanshahi et al. have assessed the effects of DBS on executive function using that test. Accordingly, the meaning of the 'disinhibition side-effect profile' keeps obscure for me. The authors should fully address this concern.

This topic has been extended on the corresponding section. It has been demonstrated on several studies using inhibitory control tests (such as the Stroop test) that DBS interferes with response

inhibition (Frank et al. 2007, Witt et al. 2004, Ray et al. 2009). Response inhibition is a fundamental cognitive function, and its impairment could lead to impulsive conduct (Bari et al. 2013). For that reason, it is possible that stimulation-related impulsivity is the result of an impairment of response inhibition caused by DBS. This could be explained as beta oscillations play a significant role in a physiological response inhibition (Brittain et al, 2012). For that reason, stimulation-induced impulsivity could be mediated by an excessive stimulation-mediated beta suppression. The pathophysiology of endogenous impulse-control disorder could be therefore different than the impulsivity mediated by stimulation, and for that reason differences could be not observed between ICD patients and control patients on inhibitory control tests. The text in the section “Objectives” has been also changed to: “response inhibition side-effect profile”.